# Two locus inheritance of non-syndromic midline craniosynostosis via rare *SMAD6* and common *BMP2* alleles

Andrew T Timberlake[1,2,3], Jungmin Choi[1,2], Samir Zaidi[1,2], Qiongshi Lu[4], Carol Nelson-Williams[1,2], Eric D Brooks[3], Kaya Bilguvar[1,5], Irina Tikhonova[5], Shrikant Mane[1,5], Jenny F Yang[3], Rajendra Sawh-Martinez[3], Sarah Persing[3], Elizabeth G Zellner[3], Erin Loring[1,2,5], Carolyn Chuang[3], Amy Galm[6], Peter W Hashim[3], Derek M Steinbacher[3], Michael L DiLuna[7], Charles C Duncan[7], Kevin A Pelphrey[8], Hongyu Zhao[4], John A Persing[3], Richard P Lifton[1,2,5,9]*

[1]Department of Genetics, Yale University School of Medicine, New Haven, United States; [2]Howard Hughes Medical Institute, Yale University School of Medicine, New Haven, United States; [3]Section of Plastic and Reconstructive Surgery, Department of Surgery, Yale University School of Medicine, New Haven, United States; [4]Department of Biostatistics, Yale University School of Medicine, New Haven, United States; [5]Yale Center for Genome Analysis, New Haven, United States; [6]Craniosynostosis and Positional Plagiocephaly Support, New York, United States; [7]Department of Neurosurgery, Yale University School of Medicine, New Haven, United States; [8]Child Study Center, Yale University School of Medicine, New Haven, United States; [9]The Rockefeller University, New York, United States

*For correspondence: richard. lifton@yale.edu

Competing interests: The authors declare that no competing interests exist.

**Abstract** Premature fusion of the cranial sutures (craniosynostosis), affecting 1 in 2000 newborns, is treated surgically in infancy to prevent adverse neurologic outcomes. To identify mutations contributing to common non-syndromic midline (sagittal and metopic) craniosynostosis, we performed exome sequencing of 132 parent-offspring trios and 59 additional probands. Thirteen probands (7%) had damaging de novo or rare transmitted mutations in *SMAD6*, an inhibitor of BMP – induced osteoblast differentiation ($p<10^{-20}$). *SMAD6* mutations nonetheless showed striking incomplete penetrance (<60%). Genotypes of a common variant near *BMP2* that is strongly associated with midline craniosynostosis explained nearly all the phenotypic variation in these kindreds, with highly significant evidence of genetic interaction between these loci via both association and analysis of linkage. This epistatic interaction of rare and common variants defines the most frequent cause of midline craniosynostosis and has implications for the genetic basis of other diseases.

## Introduction

The cranial sutures are not fused at birth, allowing for doubling of brain volume in the first year of life and continued growth through adolescence (*Persing et al., 1989*). The metopic suture normally closes between 6 and 12 months, while the sagittal, coronal, and lambdoid sutures typically fuse in adulthood (*Persing et al., 1989*; *Weinzweig et al., 2003*). Premature fusion of any of these sutures can result in brain compression and suture-specific craniofacial dysmorphism (*Figure 1*). Studies of syndromic forms of craniosynostosis, each with prevalence of ~1/60,000 to 1/1,000,000 live births and collectively accounting for 15–20% of all cases, have implicated mutations in more than 50 genes

**eLife digest** The bones in the front, back and sides of the human skull are not fused to one another at birth in order to allow the brain to double in size during the first year of life and continue growing into adulthood. However, one in 2,000 infants is born with a condition called craniosynostosis in which some of these bones have already fused. This fusion prevents the skull from growing properly, and can lead to the brain becoming compressed. As such, surgeons routinely undo the fusion in these infants to allow the brain and skull to grow normally.

Eighty-five percent of craniosynostosis cases occur in infants with no other abnormalities, (called non-syndromic cases) and most have no other affected family member. It has therefore been unclear whether these infants have craniosynostosis due to a genetic or non-genetic cause. If the cause is genetic, it is also not clear whether a mutation in a single gene, the combined effects of many genes, or something in between is responsible.

Now, by focusing on a group of 191 infants with premature fusion of bones joined at the midline of the skull, Timberlake et al. asked if any of the approximately 20,000 genes in the human genome were altered more frequently in these infants than would be expected by chance. This search revealed that rare mutations that disable one copy of a gene called *SMAD6* in combination with a common DNA variant near another gene called *BMP2* account for about 7% of infants with midline forms of craniosynostosis. These genes are both known to regulate how bones form, which explains how the mutation of these genes could lead to craniosynostosis.

In all cases, the parents of these children were unaffected. This was typically because one parent had only the *SMAD6* mutation while the other had only the common *BMP2* variant; the transmission of both to their offspring resulted in craniosynostosis. The finding that a rare mutation's effect is strongly modified by a common variant from another site in the genome is unprecedented. These findings will allow doctors to counsel families about the risk of having additional children with craniosynostosis.

Timberlake et al. next plan to study more patients with craniosynostosis to identify additional genes that contribute to this disease. They will also look at other diseases to see whether the combination of rare mutation and common DNA variant could be behind other unexplained disorders.

(*Twigg and Wilkie, 2015*; *Flaherty et al., 2016*). For example, mutations that increase MAPK/ERK signaling (e.g. *FGFR1-3* (*Twigg and Wilkie, 2015*; *Flaherty et al., 2016*), *ERF* [*Twigg et al., 2013*]) cause rare syndromic coronal or multisuture craniosynostosis, while mutations that perturb SMAD signaling (e.g. *TGFBR1/2* [*Loeys et al., 2005*], *SKI* [*Doyle et al., 2012*], *RUNX2* [*Mefford et al., 2010*; *Javed et al., 2008*]) cause rare syndromes involving the midline (sagittal and metopic) sutures. While the detailed pathophysiology of premature suture fusion has not been elucidated, aberrant signaling in cranial neural crest cells during craniofacial development has been suggested as a common mechanism (*Mishina and Snider, 2014*; *Komatsu et al., 2013*).

Despite success in identifying the genes underlying rare syndromic craniosynostosis, mutations in these genes are very rarely found in their non-syndromic counterparts (*Boyadjiev and International Craniosynostosis Consortium, 2007*). Non-syndromic craniosynostosis of the midline sutures account for 50% of all craniosynostosis (*Slater et al., 2008*; *Greenwood et al., 2014*). A GWAS of non-syndromic sagittal craniosynostosis has implicated common variants in a segment of a gene desert ~345 kb downstream of *BMP2*, and within an intron of *BBS9*; these risk alleles have unusually large effect (odds ratios > 4 at each locus) (*Justice et al., 2012*). Nonetheless, rare alleles with large effect have not been identified to date in non-syndromic sagittal or metopic craniosynostosis. We considered that the often sporadic occurrence of non-syndromic craniosynostosis might frequently be attributable to de novo mutation or incomplete penetrance of rare transmitted variants.

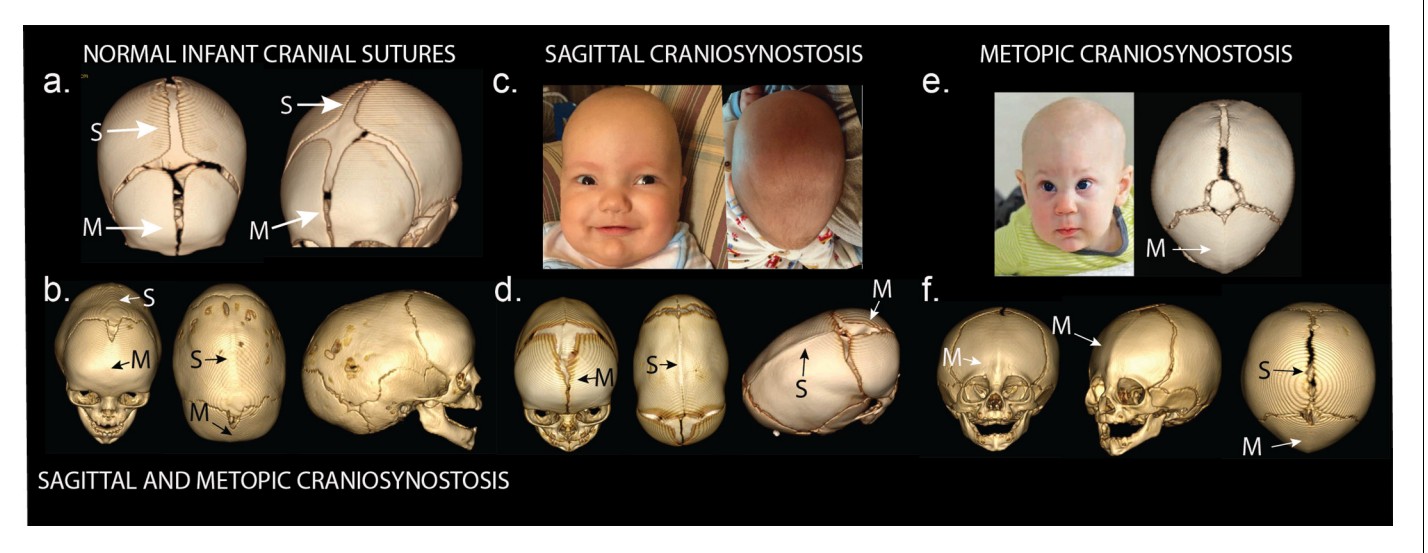

**Figure 1.** Phenotypes of midline craniosynostosis. (a) Normal infant skull with patent sagittal (S) and metopic (M) sutures. (b) Three-dimensional reconstruction of computed tomography (3D CT) demonstrating premature fusion of both the sagittal and metopic sutures. (c) A three-month-old boy with sagittal craniosynostosis featuring scaphocephaly (narrow and elongated cranial vault), and frontal bossing. (d) 3D CT reconstruction of a one-month-old boy found to have sagittal craniosynostosis. (e) A six-month-old boy presenting with trigonocephaly (triangulation of the cranial vault, with prominent forehead ridge resulting from premature fusion of the metopic suture) and hypotelorism (abnormally decreased intercanthal distance, also a result of premature fusion of the metopic suture). 3D CT reconstruction demonstrated metopic craniosynostosis. (f) 3D CT reconstruction demonstrating premature fusion of the metopic suture with characteristic trigonocephaly and hypotelorism.

## Results

### Exome sequencing of non-syndromic midline craniosynostosis

We recruited a cohort of 191 probands with non-syndromic midline craniosynostosis, including 132 parent-offspring trios and 59 probands with one parent, along with selected extended family members (see Materials and methods). All probands had undergone reconstructive surgery for either isolated sagittal (n = 113), metopic (n = 70) or combined sagittal and metopic (n = 8) craniosynostosis. Seventeen kindreds had 1 to 3 additional affected family members, including 7 parents, 12 siblings, 3 aunts/uncles and 4 more distant relatives of probands. DNA was prepared from buccal swab samples. Exome sequencing was performed as described in Materials and methods; summary data are shown in *Supplementary file 1A*. Variants were called using the GATK pipeline (see Materials and methods) and de novo mutations in parent-offspring trios were called using TrioDeNovo (*Wei et al., 2015*). The impact of identified missense variants on protein function was inferred using MetaSVM (*Dong et al., 2015*). All de novo calls were verified by in silico visualization of aligned reads (*Figure 2—figure supplement 1*), and all calls contributing to significant results for individual genes were verified by direct Sanger sequencing.

We identified a total of 144 de novo mutations, providing a de novo mutation rate of $1.64 \times 10^{-8}$ per base pair, and 1.09 de novo mutations in the coding region per offspring, consistent with prior experimental results and expectation (*Ware et al., 2015*; *Homsy et al., 2015*) (*Table 1*). Comparison of the observed distribution of mutation types to the expected from the Poisson distribution demonstrated significant enrichment of protein-altering mutations, predominantly accounted for by an excess of damaging missense mutations (MetaSVM D-mis; 28 observed D-mis compared to 14.5 expected, $p=1.0 \times 10^{-3}$, 1.93-fold enrichment), with a corresponding paucity of silent mutations (21 compared to 40.4 expected, $p=3.0 \times 10^{-4}$). From the difference in the observed vs. expected number of de novo protein-altering mutations per subject, we infer that these de novo mutations contribute to ~15% of non-syndromic midline craniosynostosis.

**Table 1.** Enrichment of protein-altering de novo mutations in 132 subjects with sagittal and/or metopic craniosynostosis.

| | Observed | | Expected | | Enrichment | p-value |
|---|---|---|---|---|---|---|
| Class | # | #/subject | # | #/subject | | |
| All mutations | 144 | 1.09 | 142.8 | 1.08 | 1.01 | 0.47 |
| Synonymous | 21 | 0.16 | 40.4 | 0.31 | 0.52 | $3.0 \times 10^{-4}$ |
| Protein altering | 123 | 0.93 | 102.4 | 0.78 | 1.17 | 0.03 |
| Total missense | 110 | 0.83 | 89.7 | 0.68 | 1.23 | 0.02 |
| T-mis | 82 | 0.62 | 75.2 | 0.57 | 1.09 | 0.23 |
| D-mis | 28 | 0.21 | 14.5 | 0.11 | 1.93 | $1.0 \times 10^{-3}$ |
| Loss of function (LOF) | 13 | 0.10 | 12.7 | 0.10 | 1.03 | 0.50 |
| LOF + D-mis | 41 | 0.31 | 27.1 | 0.21 | 1.51 | $7.8 \times 10^{-3}$ |

#, number of de novo mutations in 132 subjects; #/subject, number of de novo mutations per subject; Damaging and tolerated missense called by MetaSVM (D-mis, T-mis respectively); Loss of function denotes premature termination, frameshift, or splice site mutation. For mutation classes with enrichment compared to expectation, p-values represent the upper tail of the Poisson probability density function. For mutation classes in which we observed a paucity of mutations compared to expectation, p-values represent the lower tail.

Source data 1. De novo mutations in 132 trios with sagittal and/or metopic craniosynostosis. Mutations highlighted in orange are likely loss of function mutations, those highlighted in blue are likely damaging missense mutations (D-mis) as called by MetaSVM, and those without highlight are predicted to be tolerated (T-mis) or are synonymous (syn).

## De novo and transmitted mutations in *SMAD6*

Analysis of de novo mutation burden revealed that a single gene, *SMAD6*, harbored three de novo mutations, including two inferred loss of function (LOF) mutations (p.Q78fs*41 and p.E374*) and one D-mis mutation (p.G390C). All three were heterozygous and occurred in families in which the proband was the sole affected member. Two de novo mutations occurred in probands and one occurred in an unaffected mother of a proband (*Figure 2*). All three de novo mutations were confirmed by Sanger sequencing of PCR amplicons containing the putative mutation (*Figure 2—figure supplement 2*). *TTN*, which encodes the largest human protein, was the only other gene with more than one protein altering de novo mutation, and both of these were predicted by MetaSVM to encode tolerated variants (p.I3580M, p.T19373S). From the prior probability of de novo mutation of each base in *SMAD6* and the impact on the encoded protein (*Samocha et al., 2014*), the probability of seeing at least two de novo LOFs and one missense mutation by chance in a cohort of this size was $3.6 \times 10^{-9}$ (*Table 2*). Similarly, observing two or more de novo LOF mutations in any gene in this cohort was not expected by chance (p=$8.4 \times 10^{-3}$, see Materials and methods). Lastly, *SMAD6* is not unusually mutable, as we found no de novo *SMAD6* mutations in 900 control trios comprising healthy siblings of individuals with autism (*Iossifov et al., 2014*; *O'Roak et al., 2011*; *Sanders et al., 2012*). These findings provide highly significant evidence implicating damaging mutations in *SMAD6* as a cause of midline suture craniosynostosis.

We next considered the total burden of rare (prospectively specified allele frequency in ExAC database <$2 \times 10^{-5}$) LOF and D-mis mutations in each gene in probands. Among 191 probands, we found 1135 rare LOF and 3156 rare damaging (LOF + D-mis) alleles. The probability of the observed number of rare variants in each gene occurring by chance was calculated from the binomial distribution after adjusting for the length of each gene; Q-Q plots comparing the observed and expected P-value distributions are shown in *Figure 3*. The observed distribution conforms closely to expected with the exception of *SMAD6*. The expected number of rare LOF alleles in *SMAD6* in probands was 0.05, and the observed number was 8 (p=$1.1 \times 10^{-15}$, 156-fold enrichment). Similarly, there were 13 rare damaging variants in *SMAD6* compared to 0.14 expected (p=$1.3 \times 10^{-21}$, 91.4-fold enrichment). All of these *SMAD6* variants were confirmed by direct Sanger sequencing (*Figure 2—figure supplement 2*). All were heterozygous and different from one another (*Figure 2—source data 1*); 11 were absent among >$10^5$ alleles in the ExAC database, while two were previously seen, each once in ExAC (p.E407* and p.R465C, ExAC allele frequencies $9.0 \times 10^{-6}$ and $9.4 \times 10^{-6}$ respectively). The

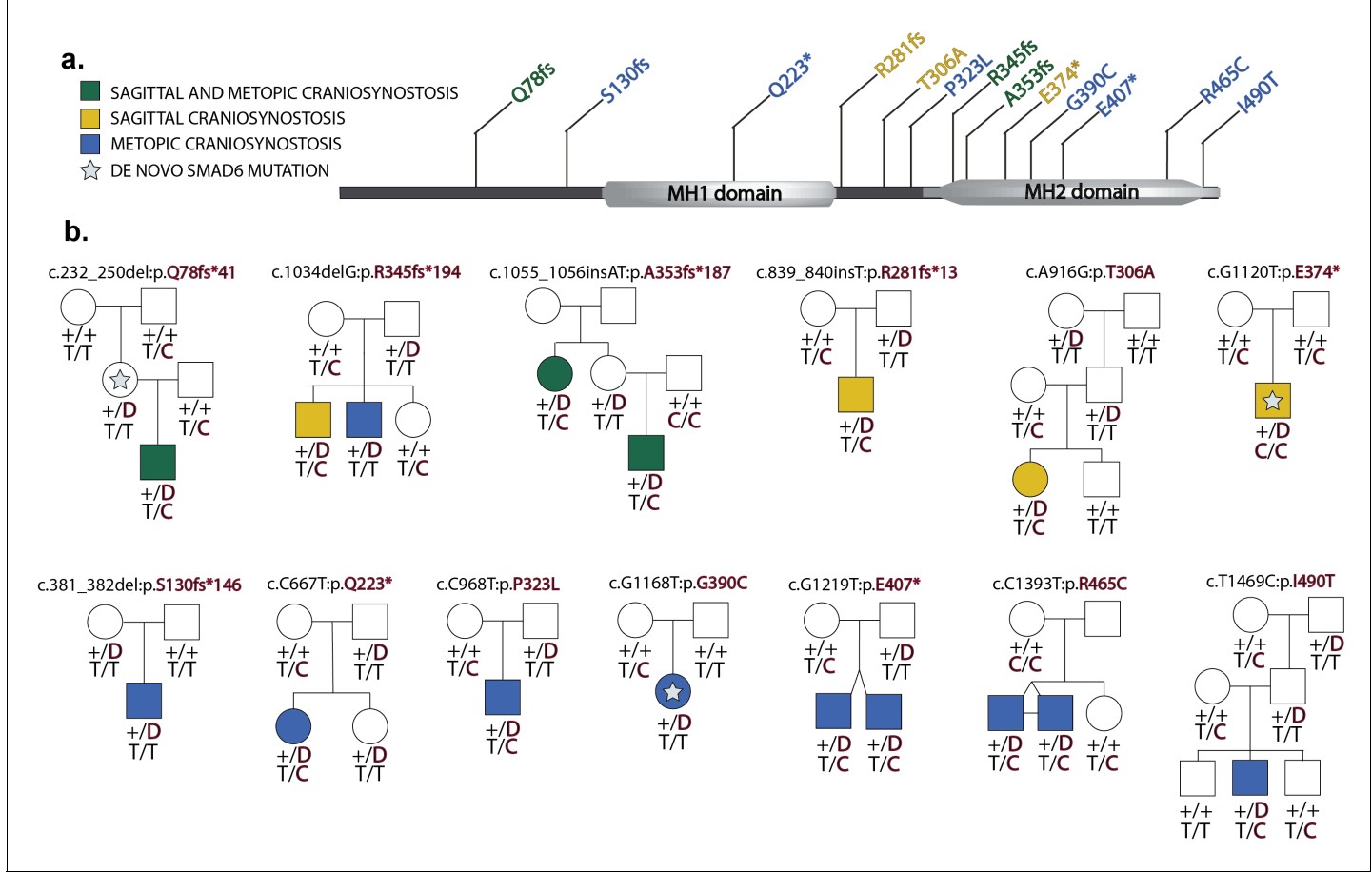

**Figure 2.** Segregation of *SMAD6* mutations and *BMP2* SNP genotypes in pedigrees with midline craniosynostosis. (a) Domain structure of *SMAD6* showing location of the MH1 and MH2 domains. The MH1 domain mediates DNA binding and negatively regulates the functions of the MH2 domain, while the MH2 domain is responsible for transactivation and mediates phosphorylation-triggered heteromeric assembly with receptor SMADs. De novo or rare damaging mutations identified in craniosynostosis probands are indicated. Color of text denotes suture(s) showing premature closure. (b) Pedigrees harboring de novo (denoted by stars within pedigree symbols) or rare transmitted variants in *SMAD6*. Filled and unfilled symbols denote individuals with and without craniosynostosis, respectively. The *SMAD6* mutation identified in each kindred is noted above each pedigree. Below each symbol, genotypes are shown first for *SMAD6* (with 'D' denoting the damaging allele) and for rs1884302 risk locus downstream of *BMP2*, (with 'T' conferring protection from and 'C' conferring increased risk of craniosynostosis). All 17 subjects with craniosynostosis have *SMAD6* mutations, and 14/17 have also inherited the risk allele at rs1884302, whereas only 3 of 16 *SMAD6* mutation carriers without the rs1884302 risk allele have craniosynostosis.

The following source data and figure supplements are available for figure 2:

**Source data 1.** Variants identified in *SMAD6*.

**Source data 2.** PCR primer sequences for Sanger sequencing of reported variants.

**Figure supplement 1.** Plots of independent Illumina sequencing reads in a parent-offspring trio showing de novo *SMAD6* mutation.

**Figure supplement 2.** Confirmation of *SMAD6* mutations by Sanger sequencing of PCR products.

results for *SMAD6* remain highly significant after excluding de novo mutations and only analyzing transmitted variants (*Figure 3—figure supplement 1*), demonstrating a significant contribution of both de novo and transmitted variants (*Table 3*). The fact that eight of the 13 rare heterozygous damaging variants in *SMAD6* seen in our cohort are frameshift (n = 5) or premature termination (n = 3) mutations, which are distributed throughout the encoded protein (*Figure 2a*), strongly supports haploinsufficiency as the mechanism of the genetic contribution of *SMAD6* to craniosynostosis.

**Table 2.** Probability of observed de novo mutations in *SMAD6* and Sprouty genes occurring by chance in 132 subjects using gene-specific mutation probabilities.

| Gene(s) | Mutations | Number of observed mutations | Number of expected mutations | p value |
|---|---|---|---|---|
| *SMAD6* | Loss of function | 2 | 0.00026 | $3.31 \times 10^{-8}$ |
| *SMAD6* | Missense | 1 | 0.0046 | $4.67 \times 10^{-3}$ |
| *SPRY1, SPRY2, SPRY3, SPRY4* | Nonsense, splice site, frameshift | 2 | 0.001193 | $7.11 \times 10^{-7}$ |

Probabilities calculated from the Poisson distribution using DenovolyzeR. The probability of observing at least 2 LOF and 1 missense mutation in *SMAD6* was $3.6 \times 10^{-9}$ via Fisher's method.

Lastly, we compared the frequency of rare (allele frequency $<2 \times 10^{-5}$ in the ExAC database) damaging (LOF + D-mis) variants in all genes in 172 European probands and 3337 unrelated European controls, who were parents of autism probands sequenced to a similar depth of coverage and

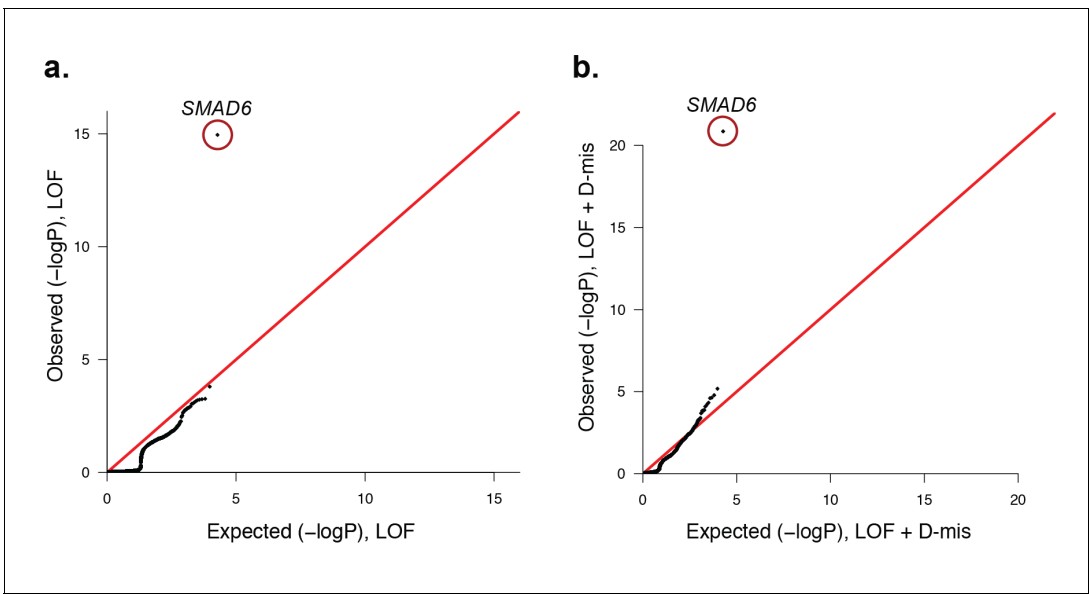

**Figure 3.** Quantile-quantile plots of observed versus expected p-values comparing the burden of rare LOF and damaging (LOF + D-mis) variants in protein-coding genes in craniosynostosis cases. Rare (allele frequency $<2 \times 10^{-5}$ in the ExAC03 database) loss of function (LOF) and damaging missense (D-mis) variants were identified in 191 probands. The probability of the observed number of variants in each gene occurring by chance was calculated from the total number of observed variants and the length of the coding region of each gene using the binomial test. The distribution of observed P-values compared to the expected distribution is shown. (**a**) Q-Q plot for rare LOF variants in each gene from a total of 1135 LOF variants identified in probands. The distribution of observed p-values closely conforms to expectation with the exception of *SMAD6*, which shows p=1.1 × $10^{-15}$ and 156-fold enrichment in cases. (**b**) Q-Q plot for rare damaging (LOF + D-mis) variants in each gene from a total of 3156 damaging variants in probands. Again, *SMAD6* deviates greatly from the expected distribution, with p<$10^{-20}$ and 91-fold enrichment.

The following source data and figure supplements are available for figure 3:

**Source data 1.** Source data for *Figure 3—figure supplement 3*.

**Figure supplement 1.** Quantile-quantile plots comparing all transmitted, damaging variants in protein-coding genes in 191 probands with midline craniosynostosis to the expected binomial distribution.

**Figure supplement 2.** Principal-component analysis of 191 probands and 3337 European autism controls.

**Figure supplement 3.** Quantile-quantile plot of observed versus expected p-values comparing the burden of damaging (LOF + D-mis) variants in protein-coding genes in craniosynostosis cases and controls.

**Table 3.** Enrichment of de novo and transmitted damaging variants in *SMAD6* in craniosynostosis.

| | Observed | Expected | Enrichment | p-value |
|---|---|---|---|---|
| De novo LOF and D-mis | 3 | 0.0049 | 612 | $3.6 \times 10^{-9}$ |
| Transmitted LOF and D-mis | 10 | 0.1404 | 71.2 | $7.0 \times 10^{-16}$ |
| Total | 13 | 0.1453 | 89.5 | $1.4 \times 10^{-22}$ |

LOF, loss of function; D-mis, damaging missense variants per MetaSVM; The total number of *SMAD6* variants expected in this cohort was calculated by summing the expected number of de novo and transmitted variants. P-value combining probabilities from de novo and transmitted protein damaging *SMAD6* variants was determined by Fisher's method.

analyzed in a similar fashion (see Materials and methods, *Supplementary file 1A*). European ancestry was determined by principal component analysis of exome sequence data (*Figure 3—figure supplement 2*). Q-Q plots showed that the observed distribution of Fisher's exact statistics comparing the frequency of damaging variants in cases and controls closely corresponded to the expected distribution, again with the exception of *SMAD6,* in which cases showed enrichment of damaging variants (p=$6.3 \times 10^{-8}$) and LOF variants (p=$5.7 \times 10^{-6}$) (*Figure 3—figure supplement 3*). Significant enrichment was also seen in comparison to European NHLBI and ExAC controls (*Figure 3—source data 1*). The odds ratios for association of all damaging variants in *SMAD6* in cases vs. controls was consistent across control cohorts, ranging from 26.9 to 35.1; the odds ratios for LOFs ranged from 102.6 to infinity (owing to zero LOF's in autism controls).

Aside from *SMAD6*, no other single gene approached genome-wide significance in these analyses of dominant alleles. Analysis of recessive genotypes, considering alleles with frequency <$10^{-3}$, identified no genes with more than one rare recessive genotype.

Collectively, the significant burden of both de novo and rare transmitted mutations, along with significant association results in case-control analysis provide extremely strong evidence that rare damaging *SMAD6* alleles impart large effects on risk of non-syndromic midline craniosynostosis.

*SMAD6* mutations were significantly more frequent in kindreds with any metopic craniosynostosis (10 of 78, 12.8%) compared to those with isolated sagittal craniosynostosis (3 of 113, 2.7%; p=$8.1 \times 10^{-3}$ by Fisher's exact test, odds ratio 5.3). These results suggest that mutations in *SMAD6* confer greater risk for metopic suture closure. We found no significant correlation between the type of mutation (LOF vs. D-mis) or location within the gene of *SMAD6* mutation and phenotypic class (*Table 4*).

Interestingly, transmitted *SMAD6* mutations were significantly enriched in kindreds with familial craniosynostosis, accounting for 4 of 17 kindreds with more than one affected subject (p=0.02, Fisher's exact test; odds ratio 5.6). In these kindreds, all four additional affected subjects carried the *SMAD6* mutation found in the proband (*Figure 2*).

SMAD6 is a member of the inhibitory-SMAD family. Activation of BMP receptors leads to phosphorylation of receptor SMADs, which can complex with SMAD4, translocate to the nucleus and partner with RUNX2 to induce transcription of genes that promote osteoblast differentiation (*Javed et al., 2008*; *Hata et al., 1998*) (*Figure 4a*). This process is inhibited by SMAD6 binding to phosphorylated receptor SMADs, forming an inactive complex. SMAD6 also inhibits BMP signaling

**Table 4.** Distribution of suture involvement in kindreds with and without rare (allele frequency < $2 \times 10^{-5}$) de novo and transmitted damaging (LOF + D-mis) variants in *SMAD6*.

| | Total # kindreds | Total # *SMAD6* mutations (%) | # LOF (%) |
|---|---|---|---|
| Sagittal | 113 | 3 (2.7) | 2 (1.8) |
| Metopic | 70 | 7 (10) | 3 (3.9) |
| Sagittal and Metopic | 8 | 3 (37.5) | 3 (37.5) |
| Total | 191 | 13 (6.8) | 8 (4.2) |

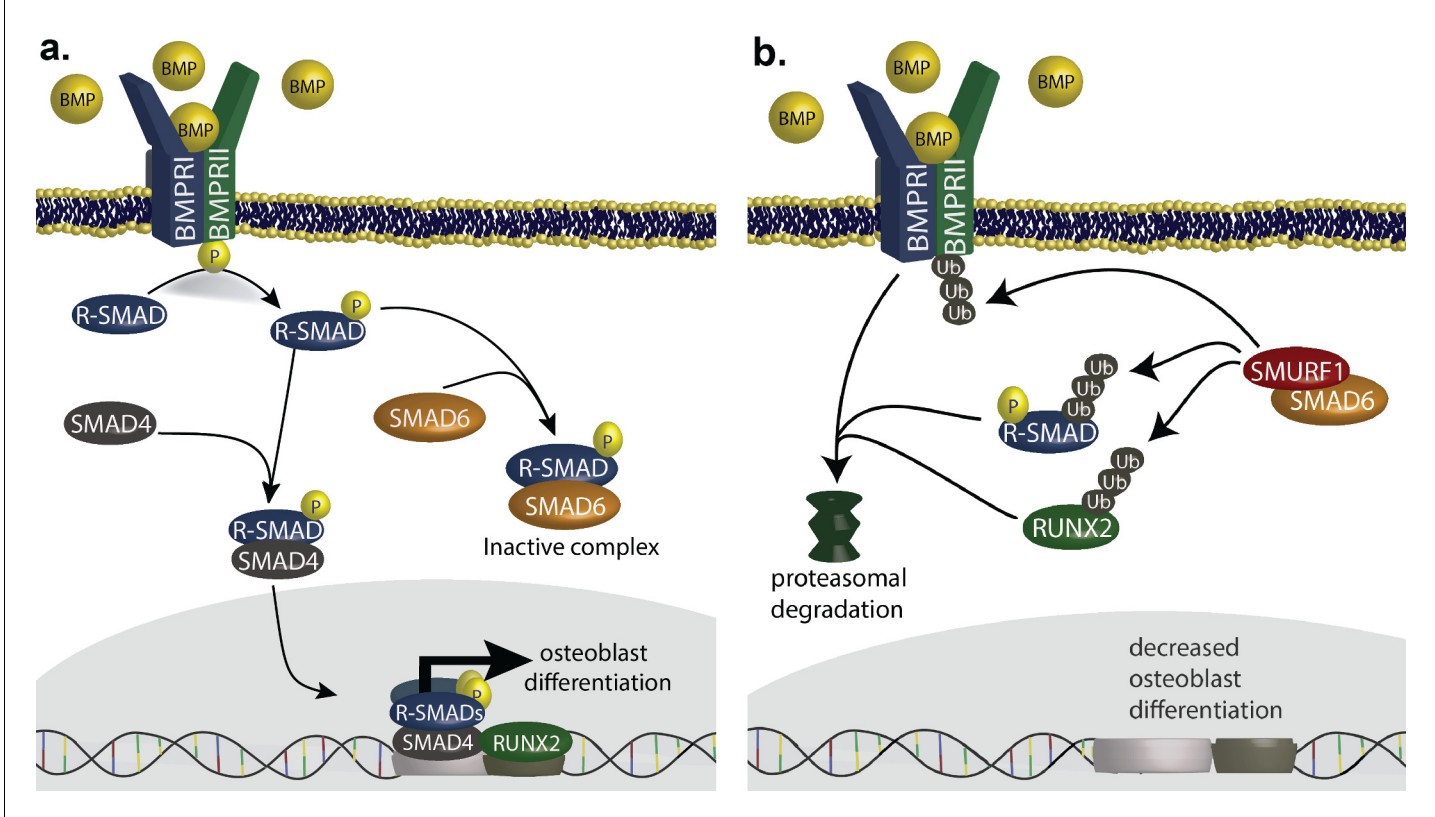

**Figure 4.** SMAD6 inhibits osteoblast differentiation by inhibiting BMP-mediated SMAD signaling (*Salazar et al., 2016*). (a) BMP ligands activate BMP receptors, leading to phosphorylation of receptor-regulated SMADs (R-SMADs), which complex with SMAD4 and enter the nucleus, cooperating with RUNX2 to induce osteoblast differentiation. SMAD6 inhibits this signal by competing with SMAD4 for binding to R-SMADs, preventing nuclear translocation. (b) SMAD6 also cooperates with SMURF1, an E3 ubiquitin ligase, to induce ubiquitin-mediated proteasomal degradation of R-SMADs, BMP receptor complexes, and RUNX2.

by complexing with the ubiquitin ligase SMURF1, which ubiquitylates BMP receptors, receptor SMADs and RUNX2, leading to their proteasomal degradation (*Figure 4b*) (*Murakami et al., 2003*). This pathway plays a well-established role in the development of the cranial vault and closure of cranial sutures. In mice, constitutive activity of BMPR1A in cranial neural crest results in SMAD-dependent development of metopic craniosynostosis (*Komatsu et al., 2013*), and genetic deficiency for the SMAD inhibitor *SMURF1* causes midline craniosynostosis (*Shimazu et al., 2016*). Similarly, duplication of *RUNX2* causes syndromic metopic craniosynostosis in humans (*Mefford et al., 2010*). Lastly, *SMAD6* knockout mice are born with domed skulls and show anomalous bone deposition in the metopic suture; they also show an augmented and prolonged response to BMP2 stimulation (*Estrada et al., 2011*; *Retting, 2008*). These findings are consistent with haploinsufficiency as the mechanism of *SMAD6* mutations in craniosynostosis, with loss of the inhibitory effect of SMAD6 promoting increased BMP signaling and premature closure of sutures.

We explored our kindreds for other mutations in this signaling pathway. Interestingly, we identified one de novo D-mis mutation in *SMURF1* in a proband with sporadic metopic craniosynostosis (*Figure 5*).

## Incomplete penetrance of *SMAD6* mutations explained by a common variant near *BMP2*

Within the 13 kindreds harboring rare damaging *SMAD6* variants, all 17 affected subjects had the *SMAD6* mutation found in the proband (*Figure 2*). Nonetheless, *SMAD6* mutations showed striking incomplete penetrance. In particular, zero of 10 parental *SMAD6* mutation carriers had a diagnosis

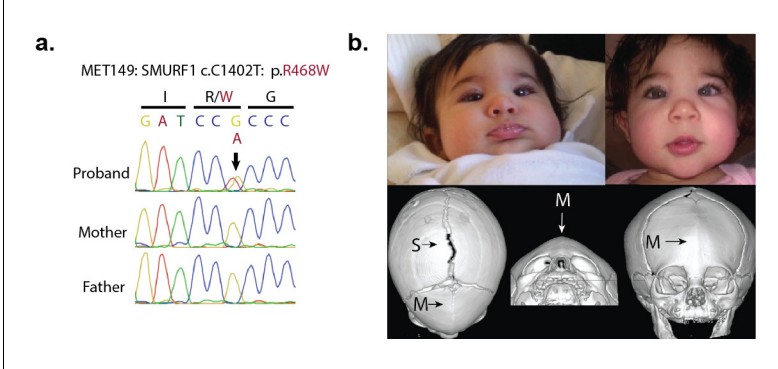

**Figure 5.** A de novo variant identified in *SMURF1*. (**a**) Sanger sequence electropherogram of a PCR product amplified from the genomic DNA of a proband with metopic craniosynostosis, confirming a de novo R468W mutation in *SMURF1*, a *SMAD6* binding partner. (**b**) Patient photographs of the proband, who presented with trigonocephaly and mild orbital abnormalities. 3D CT reconstruction demonstrates metopic craniosynostosis, trigonocephaly, and a patent sagittal suture.

of, or showed evidence of craniosynostosis. Examination of Illumina read counts and Sanger sequence traces provided no suggestion that the mutations were mosaic in unaffected parents (mean/median of 52.9%/52.4% variant reads in transmitting parents, respectively; range 37.3% to 71.4%). There was no significant effect of gender on penetrance. From the data in these kindreds, the penetrance of *SMAD6* mutations is estimated at 24% following exclusion of probands, who were ascertained for the presence of disease (57% if probands are included). The striking absence of craniosynostosis among transmitting parents suggests the possibility of purifying selection, with subjects having craniosynostosis less likely to have offspring.

We considered whether inheritance at other genetic loci might account for the striking incomplete penetrance of *SMAD6* mutations in these kindreds. A previous GWAS of non-syndromic sagittal craniosynostosis implicated common variants ~345 kb downstream of the closest gene, *BMP2* (encoding bone morphogenetic protein 2), with unusually large effect size (e.g., rs1884302, with risk allele frequency of 0.34 and odds ratio of 4.6). BMP2 is a ligand for BMP receptors upstream of SMAD signaling (*Salazar et al., 2016*) and is an inducer of osteogenesis. We posited that risk alleles at this locus might increase the penetrance of *SMAD6* mutations by increasing BMP2 levels and further increasing SMAD signaling. Genotypes for rs1884302 are shown in *Figure 2* and provide strong evidence of epistatic interaction between *SMAD6* and *BMP2* alleles. Fourteen individuals had both a *SMAD6* mutation and the rs1884302 risk allele; 100% of these had craniosynostosis. In contrast, 16 subjects had a *SMAD6* mutation but no rs1884302 risk allele; only 3 of these individuals (19%) had craniosynostosis. Lastly, 0 of 18 members of these kindreds who had only the rs1884302 risk allele had craniosynostosis. The relationship of these two genotypes to craniosynostosis in these kindreds was highly significant (p=1.4 × 10⁻¹⁰ by the Freeman-Halton extension of Fisher's exact test; *Table 5*).

Confining analysis just to subjects with *SMAD6* mutation, there was dramatically increased occurrence of craniosynostosis among those with the rs1884302 risk allele compared to those without (p=4.8 × 10⁻⁶ by Fisher's exact test). The presence of the rs1884302 risk allele increased the risk of craniosynostosis >5-fold with a very high odds ratio that includes infinity owing to 100% penetrance among those with risk genotypes at both loci. This two locus contribution is further supported by significant transmission disequilibrium, with rs1884302 risk alleles transmitted from heterozygous parents to affected offspring in 11 of 13 transmissions (p=0.013 by Chi-square, *Supplementary file 1B*). In sum, inheritance at rs1884302 explains nearly all of the variation in phenotype among subjects with *SMAD6* mutations and demonstrates two locus transmission of craniosynostosis.

In contrast, common variants at the *BBS9* locus that also showed strong association with midline craniosynostosis in case-control analysis (OR > 4) (*Justice et al., 2012*) showed no significant interaction with *SMAD6* (TDT p=0.89; *Supplementary file 1B*), demonstrating specificity of the observed interaction of rare variants in *SMAD6* and a common variant near *BMP2*.

**Table 5.** Risk of craniosynostosis in *SMAD6* mutation carriers in the presence or absence of a *BMP2* risk allele.

| SMAD6/BMP2 Genotypes | Craniosynostosis (+) | Craniosynostosis (−) |
|---|---|---|
| SMAD6 (+) / BMP2 risk allele (+) | 14 | 0 |
| SMAD6 (+) / BMP2 risk allele (−) | 3 | 13 |
| SMAD6 (−) / BMP2 risk allele (+) | 0 | 18 |

All members of kindreds found to have a mutation in *SMAD6* were included. *SMAD6*(+) indicates the presence of a heterozygous LOF or D-mis allele. The reported *BMP2* risk allele is 'C' at risk locus rs1884302, found within a gene desert ~345kb downstream of *BMP2*. $p=1.4 \times 10^{-10}$ by the Freeman-Halton extension of Fisher's exact test. Odds ratio in favor of disease was incalculable due to the absence of craniosynostosis in *SMAD6* (−) individuals in these kindreds.

Lastly, we compared the joint segregation of rare damaging *SMAD6* and common *BMP2* risk alleles to the segregation of craniosynostosis in a parametric two locus linkage model in these kindreds (see Materials and methods, *Supplementary file 1C*). The results provided extremely strong evidence supporting linkage under a two locus model, with a maximum lod score of 7.37 (odds ratio $2.3 \times 10^7$:1 in favor of linkage compared to the null hypothesis; family-specific lod scores are shown in *Supplementary file 1D*). The maximum likelihood model specified 100% penetrance of craniosynostosis when risk alleles at both loci are present, 9% penetrance when only a damaging *SMAD6* allele is present, 0.08% or 0.32% penetrance when only one or two *BMP2* risk alleles are present, and a 0.02% phenocopy rate, with zero recombination between trait and both marker loci. This two locus model was 1410 – fold more likely than than the best single locus model, in which damaging *SMAD6* variants had penetrance of 20%. These results provide extremely strong statistical support for the two locus model by linkage and extends the genetic evidence beyond simple association to linkage within pedigrees, which is not susceptible to potential confounders such as population stratification, and is insensitive to misspecification of allele frequency.

## Mutations in MAP kinase regulators

Previous research has implicated increased activity of the MAP kinase/ERK pathway in craniosynostosis (*Twigg et al., 2013*; *Shukla et al., 2007*). We identified one de novo LOF in both *SPRY1* and *SPRY4,* developmental regulators of the MAP kinase/ERK pathway; these variants comprised two of only 11 de novo LOFs other than those in *SMAD6*. The *SPRY4* mutation (p.E160*) arose de novo in a proband with sagittal craniosynostosis and no family history (*Figure 6a*). The *SPRY1* mutation (p. Q6fs*8) was de novo in a woman with mild cranial dysmorphism who did not undergo surgery, and was transmitted to both of her children, who both had sagittal craniosynostosis (*Figure 6b*). The probability of observing two or more de novo LOF mutations in any of the 4 Sprouty genes by chance in this cohort surpassed genome-wide significance ($p=7.1 \times 10^{-7}$, *Table 2*). Consistent with a role for *SPRY1* haploinsufficiency, a de novo microdeletion that included *SPRY1* has previously been reported in a child with sagittal craniosynostosis (*Fernández-Jaén et al., 2014*). Moreover, in mice with *TWIST1* haploinsufficiency, a model of syndromic craniosynostosis, overexpression of *SPRY1* prevents suture fusion (*Connerney et al., 2008*). Lastly, protein altering de novo mutations were also identified in other regulators and mediators of MAP kinase signaling, including *RASAL2*, *DUSP5*, *MAP3K8*, *KSR2*, *RPS6KA4*, and *RGS3*. 5 of 6 occurred in probands with sagittal craniosynostosis (*Table 1—source data 1*). Determining the significance of these findings will require further study.

## Discussion

These findings implicate a two locus model of inheritance in non-syndromic midline craniosynostosis via epistatic interactions of rare heterozygous *SMAD6* mutations and common risk alleles near *BMP2*. There is extremely strong evidence implicating each locus independently, along with highly significant evidence from both analysis of association and linkage that the risk of craniosynostosis is

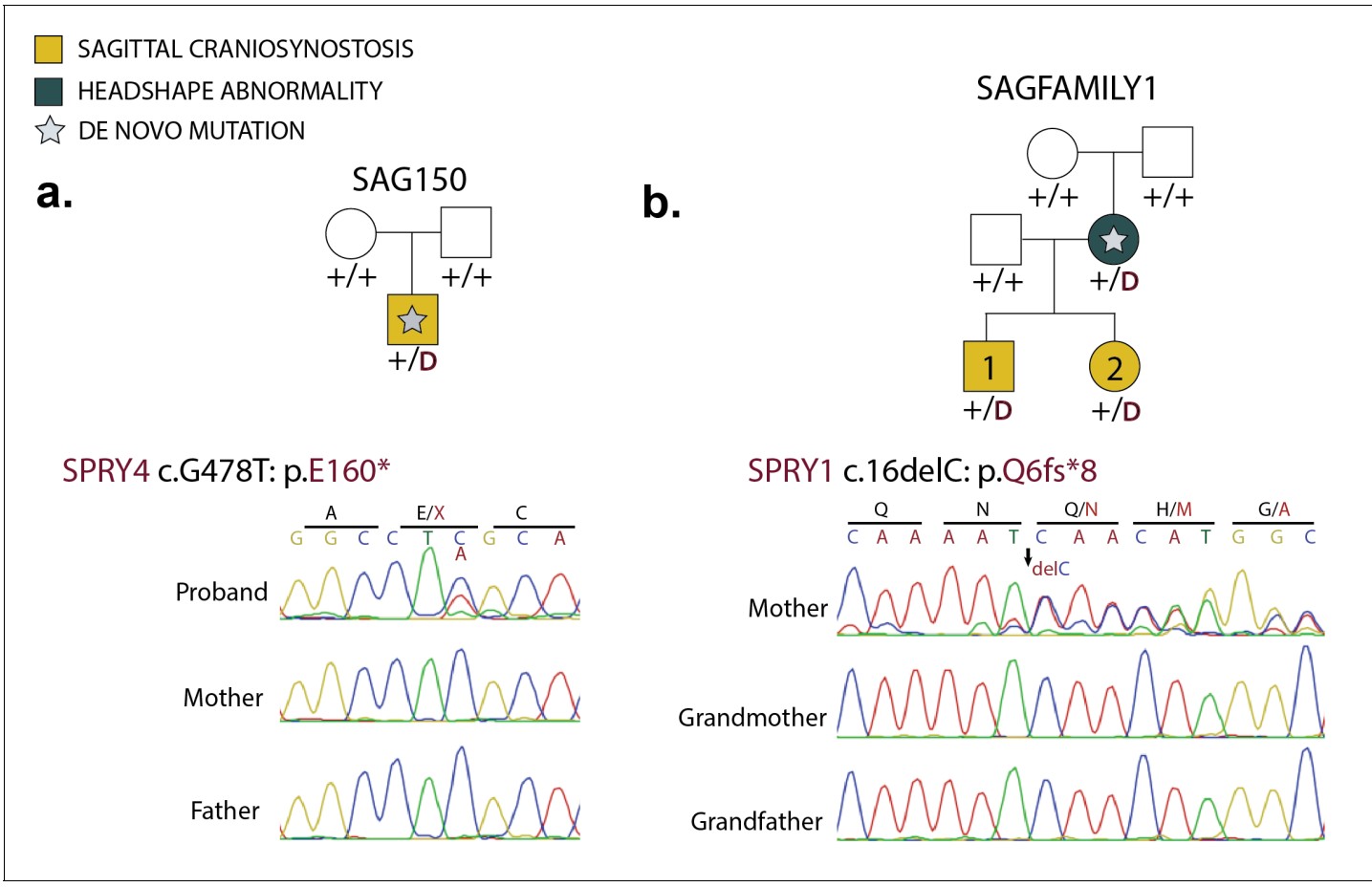

**Figure 6.** De novo loss-of-function mutations in Sprouty genes. (**a**) Pedigree and Sanger sequencing traces for kindred SAG150, demonstrating a de novo nonsense mutation in *SPRY4* (p.E160*) in the proband. (**b**) Pedigree and Sanger sequencing traces in a kindred with a de novo *SPRY1* frameshift mutation (p.Q6fs*8) that was transmitted to two affected offspring.

markedly increased in individuals carrying risk alleles at both loci compared to those with only a single risk allele at either locus. Rare damaging variants in *SMAD6* alone impart very large effects on disease risk with low penetrance, with inheritance at *BMP2* explaining nearly all of the variation in occurrence of craniosynostosis seen among *SMAD6* mutation carriers. The results support a threshold effect model, with quantitative increases in SMAD signaling resulting from reduced inhibition of SMAD signaling by SMAD6, owing to haploinsufficiency (strongly supported by a plethora of LOF variants distributed across *SMAD6*), along with a putative increase in SMAD signaling owing to increased BMP2 expression via the risk SNP rs1884302 leading to accelerated closure of midline sutures. Consistent with this model, as articulated previously, substantial prior evidence in mouse and human has implicated BMP signaling via SMADs in closure of the midline sutures, and *SMAD6* in inhibiting this pathway. Moreover, consistent with the common variant near *BMP2* modifying BMP2 expression, duplication of a nearby limb-specific enhancer increased BMP2 expression, leading to a Mendelian limb defect (*Dathe et al., 2009*). While the genetic data provide unequivocal support for the role of these two loci in midline craniosynostosis, and for haploinsufficiency as the mechanism of *SMAD6* contribution, further studies will be necessary to delineate the precise mechanism by which the risk genotypes cause disease.

*SMAD6* mutations with and without *BMP2* risk alleles account for ~7% of probands in this cohort of non-syndromic midline craniosynostosis. This frequency is much greater than any other genotype causing syndromic midline craniosynostosis (e.g., *TGFBR1/2*, *SKI*, *RUNX2*), which are sufficiently rare that their prevalence has not been well-established. Moreover, because non-syndromic sagittal and

metopic craniosynostosis comprise half of all craniosynostoses (*Slater et al., 2008*; *Greenwood et al., 2014*), *SMAD6/BMP2* genotypes are inferred to account for ~3.5% of all cranio-synostosis, and are likely rivaled in frequency only by mutations in *FGFR2* as the most frequent cause of all craniosynostoses (*Twigg and Wilkie, 2015*).

These findings will be of immediate utility in clinical diagnosis and genetic counseling. The combination of a rare damaging *SMAD6* mutation plus a common *BMP2* risk allele conferred 100% risk of craniosynostosis in our cohort, while those with a *SMAD6* mutation but no *BMP2* risk allele were at markedly lower risk. Interestingly, these *SMAD6* mutation-only cases thus far have all had isolated metopic craniosynostosis (*Figure 2*). Rare damaging SMAD6 mutations were found in nearly 25% of kindreds with recurrent midline craniosynostosis, and 37.5% of patients with combined sagittal and metopic craniosynostosis in our cohort. The precision of penetrance estimates and prevalence in specific disease subsets will improve with larger sample sizes.

Given the suggestion of a threshold for phenotypic effect, we also considered whether there might be additional *SMAD6* alleles that impart phenotypic effect that have a higher frequency than the $2 \times 10^{-5}$ threshold that we used. We found only one additional *SMAD6* damaging allele among parents in our cohort with allele frequency <0.001. Interestingly, this allele, p.E287K, (ExAC frequency $3.3 \times 10^{-5}$) was transmitted to a proband with sagittal and metopic craniosynostosis along with two doses of the *BMP2* risk allele, each inherited from a heterozygous parent.

Considering neurodevelopmental outcomes, 11 of 15 subjects with rare damaging *SMAD6* variants who are more than one year of age (and hence can have neurodevelopmental evaluation) had some form of developmental delay (*Supplementary file 1E*). While early surgical intervention provides the best neurological outcomes (*Patel et al., 2014*), more than a third of patients with non-syndromic midline craniosynostosis have subtle learning disability (*Magge et al., 2002*; *Shipster et al., 2003*; *Sidoti et al., 1996*). BMP signaling plays an essential role in vertebrate brain development (*Bier and De Robertis, 2015*), raising the possibility that aberrant BMP signaling could contribute to neurodevelopmental outcome independent of its effect on craniosynostosis. The only other clinical finding observed in more than one subject with *SMAD6* mutation was a congenital inguinal hernia in 3 patients (16.7%; *Supplementary file 1E*).

While *SMAD6* was the only single gene showing genome-wide significant burden of de novo mutation, de novo protein-altering mutations are estimated to contribute to ~15% of cases. This estimated fraction is similar to estimates for autism and congenital heart disease, other diseases in which large-scale studies have shown a role for de novo mutations (*Homsy et al., 2015*; *Iossifov et al., 2014*; *Sanders et al., 2012*; *Zaidi et al., 2013*; *De Rubeis et al., 2014*). Also like autism and congenital heart disease, few individual genes were implicated after sequencing modest numbers of trios, implying that de novo mutation in a large number of genes are likely to contribute to sagittal and metopic craniosynostosis. This observation strongly supports sequencing substantially larger numbers of non-syndromic patients, an approach that has proved highly productive for discovery of genes and pathways underlying autism and CHD.

Lastly, these results provide a clear example of the epistatic (non-additive) interaction of very rare mutations at one locus with a common variant at a second, unlinked locus. This observation adds to the small number of two locus phenotypes that have been defined with robust genetic data (*Lupski, 2012*), and suggest that other common variants, particularly those with relatively large effect, may combine with rare alleles at one or more loci to produce genotypes with high penetrance that together may account for a substantial fraction of disease risk.

## Materials and methods

### Subjects and samples

Participants were ascertained from either the Yale Pediatric Craniofacial Clinic or by responding to an invitation posted on the Cranio Kids- Craniosynostosis Support and Craniosynostosis-Positional Plagiocephaly Support Facebook pages. All participants or their parents provided written informed consent to participate in a study of the genetic causes of craniosynostosis in their family. Inclusion criteria included a diagnosis of sagittal and/or metopic craniosynostosis in the absence of known syndromic forms of disease by a craniofacial plastic surgeon or pediatric neurosurgeon. All probands had undergone reconstructive surgery. Participating family members provided buccal swab samples

(Isohelix SK-2S buccal swabs), craniofacial phenotype data, medical records, operative reports, and imaging studies. Written consent was obtained for publication of patient photographs. The study protocol was approved by the Yale Human Investigation Committee Institutional Review Board (IRB).

Control cohorts comprised 3337 previously studied healthy European parents of probands with autism, Europeans found in the NHLBI Exome Sequencing Project database (NHLBI), and Non-Finnish Europeans found in the Exome Aggregation Consortium v0.3 database (ExAC).

## Exome sequencing, risk allele genotyping, and analysis

DNA was prepared from buccal swab samples according to the manufacturer's protocol. Exome sequencing was performed by exon capture using the Roche MedExome or Roche V2 capture reagent followed by 74 base paired-end sequencing on the Illumina HiSeq 2000 instrument as previously described (*Zaidi et al., 2013*). Samples were barcoded then captured and sequenced in multiplex. Quality metrics are shown in *Supplementary file 1A*.

Sequence reads were aligned to the GRCh37/hg19 human reference genome using BWA-Mem. Local realignment and quality score recalibration were performed using the GATK pipeline, after which variants were called using the Genome Analysis Toolkit Haplotype Caller. A Bayesian algorithm, TrioDeNovo, was used to call de novo mutations (*Wei et al., 2015*). VQSR "PASS" variants with ExAC allele frequency ≤0.001 sequenced to a depth of 8 or greater in the proband and 12 or greater in each parent with Phred-scaled genotype likelihood scores >30 and de novo quality scores ($\log_{10}$(Bayes factor)) >6 were considered. Independent aligned reads at variant positions were visualized in silico to remove false calls (*Figure 2—figure supplement 1*). For de novo calls passing visual inspection, variants receiving the highest de novo genotype quality score (100) were deemed valid. Forty of these de novo mutations were selected at random for validation by bidirectional Sanger sequencing of the proband and both parents; 100% of these tests confirmed de novo mutation in the proband. The observed number of de novo variants identified per trio closely matched the expected Poisson distribution (*Supplementary file 1F*). All de novo variants named in the main text were confirmed by Sanger sequencing. Transmitted variants were similarly aligned and called as per above. All de novo and transmitted variants were annotated using ANNOVAR (*Wang et al., 2010*). Allele frequencies of identified variants were taken from the ExAC database. The impact of nonsynonymous variants was predicted using the MetaSVM rank score, with scores greater than 0.83357 serving as a threshold for predicting that the mutation was deleterious (MetaSVM 'D', D-mis) (*Dong et al., 2015*). For case control burden analysis of all protein coding genes, all GATK VQSR 'PASS' variants were considered.

In assessing the association of known risk alleles near *BMP2* and within *BBS9* with craniosynostosis in *SMAD6* mutation carriers, genotypes for rs1884302 and rs10262453 were determined by direct Sanger sequencing.

All mutations reported in *SMAD6, SMURF1, SPRY1*, and *SPRY4* were confirmed by direct bidirectional Sanger sequencing of the products of PCR amplification of segments containing putative mutations (*Figure 2—figure supplement 2*, *Figure 5*, *Figure 6*). PCR primers are listed in *Figure 2—source data 2*. Sanger sequencing electropherograms were manually inspected using Geneious after alignment to the reference genome sequence in GRCh37/hg19.

## Burden of de novo mutations

The observed distribution of mutation type was compared to pre-computed expected values across the exome using Poisson statistics as described (*Ware et al., 2015*; *Homsy et al., 2015*). Pre-calculated gene-specific mutation probabilities were used to determine the probability of the observed number and type of de novo mutations in *SMAD6* occurring by chance using denovolyzeR (*Ware et al., 2015*; *Samocha et al., 2014*). To assess the probability of observing 3 protein damaging mutations, P values for observing 2 de novo LOF's and one de novo missense mutation were combined using the Fisher's combined probability test. To assess the burden of de novo mutation in Sprouty genes, a gene set was curated in denovolyzeR including: *SPRY1, SPRY2, SPRY3, SPRY4*. The probability of observing 2 de novo LOF mutations in this gene set was calculated by comparing this number to expectation in denovolyzeR. The probability of observing more than one de novo LOF mutation in any gene in our cohort was determined using a permutation function- denovolyzeMultiHits() (*Ware et al., 2015*). In total, 13 de novo LOF variants were observed in our cohort. The

probability of observing >1 de novo LOF in any gene by chance with a set of 13 mutations was determined by 1 million iterations in which these 13 'hits' were sampled from all genes given the probability of de novo mutation in each (*Homsy et al., 2015*). The number of times any gene had more than one hit in an iteration was counted, and that number divided by 1 million represented the probability of observing more than one de novo LOF mutation in our cohort.

## Contribution of de novo mutation to craniosynostosis

We infer that the number of probands with protein altering de novo mutations (n = 123) in excess of expectation by chance (n = 102.4) represents the number of subjects in whom these mutations confer craniosynostosis risk (n = 20.6). Comparing this fraction to the total number of trios (20.6/132) yields the fraction of patients in our cohort in whom we expect these mutations contribute to disease: ~15.6%.

## Binomial test

The observed distribution of rare (ExAC frequency $<2 \times 10^{-5}$) LOF and D-mis alleles was compared to the expected distribution using the binomial test. The total number of LOF and D-mis alleles in 191 probands was tabulated, totaling 1135 LOF alleles and 2021 D-mis alleles (3156 total damaging alleles). The expected number of variants in each gene was calculated from the proportion of the exome comprising the coding region of each gene multiplied by the total number of alleles identified in cases. Enrichment was calculated as the number of observed mutations divided by the expected number.

## Transmission disequilibrium test

We used a transmission disequilibrium test to compare the transmission ($M_1$) and non-transmission ($M_2$) of *BMP2* and *BBS9* risk alleles (rs1884302 and rs10262453 respectively) to affected offspring in kindreds with *SMAD6* mutations. We tested for deviation from the expected transmission value of 50% by the binomial Chi-square test with 1 Df.

## Case vs. control comparison

A fisher exact test was used to compare the prevalence of LOF or LOF+D-mis variants in 172 craniosynostosis probands and 3337 controls, the latter comprising the unaffected parents of offspring with autism. Controls were exome sequenced on the Roche V2 capture reagent followed by sequencing on the Illumina platform (*Sanders et al., 2012*; *Krumm et al., 2015*; *Iossifov et al., 2012*). All control BAM files were processed with sequences aligned and variants called in parallel to aforementioned cases. Cases and controls, on average, both had ~94–95% of targeted bases read 8 or more times (*Supplementary file 1A*). We restricted cases and controls to European (CEU) ancestry using the EIGENSTRAT program, which compared single nucleotide polymorphism (SNP) genotypes from case and control subjects with individuals of known ancestry in HapMap3 (*Frazer et al., 2007*) (*Figure 3—figure supplement 2*). To avoid bias, exons analyzed were restricted to those that intersected between the Roche V2 and MedExome capture reagents. For genes with more than 1 LOF or D-mis variant in cases, aligned reads were visualized in silico at all variant positions in both cases and controls. For genes displaying p<0.005, we compared the burden of mutated alleles in cases to European subjects in the NHLBI ESP and ExAC databases. The total number of alleles evaluated per gene was taken as the median of the allele numbers reported for all positions across a gene in NHLBI and ExAC respectively (*Figure 3—source data 1*).

## Two locus linkage analysis

Parametric two locus linkage analysis was performed comparing the segregation of rare damaging *SMAD6* alleles and common *BMP2* risk alleles at rs1884302 to the segregation of craniosynostosis in kindreds harboring rare damaging *SMAD6* variants. Risk alleles at both loci were specified as showing zero recombination with underlying trait loci; risk/penetrance of each two locus genotype was estimated from their values at the maximum likelihood (*Supplementary file 1C*). A phenocopy rate of 0.02% was specified from estimates of disease prevalence and the fraction of disease attributable to risk genotypes. Likelihood ratios of the observed results occurring under the specified model vs. the alternative of chance were calculated for each kindred, converted to lod scores

(*Supplementary file 1D*) and the sum of lod scores across all kindreds was calculated. For kindreds (n = 2) with missing parental genotypes, the likelihood ratio of observed genotypes in offspring occurring under the parametric model compared to chance was estimated from ethnic-specific *BMP2* genotype frequencies and frequencies of rare damaging *SMAD6* alleles in missing parental genotypes. The likelihood ratio of the best two locus model was compared to that of the best single locus model considering only the segregation of rare damaging *SMAD6* variants.

### Kinship analysis

Kinship analysis was performed for all probands and controls using Plink. All trio structures were confirmed with parent-offspring pairs having PiHat values of 0.45–0.55.

### URLs

GATK: (https://www.broadinstitute.org/gatk); TrioDeNovo: (http://genome.sph.umich.edu/wiki/Trio-denovo); DenovolyzeR: (http://denovolyzer.org); Plink: (http://pngu.mgh.harvard.edu/~purcell/plink); MetaSVM/ANNOVAR: (http://annovar.openbioinformatics.org); NHLBI ESP: (http://evs.gs.washington.edu/EVS); ExAC03: (http://exac.broadinstitute.org); Geneious: (www.geneious.com). Isohelix Buccal Swab DNA isolation: (http://www.isohelix.com/wp-content/uploads/2015/09/BuccalFixIsoKit.pdf).

### Accession codes

Whole-exome sequencing data have been deposited in the database of Genotypes and Phenotypes (dbGaP) under accession phs000744. NCBI RefSeq accessions for all named genes are listed in *Table 1—source data 1*.

## Acknowledgements

We are enormously grateful to the study participants and their families for their invaluable role in this study, to the Cranio Kids- Craniosynostosis Support Group and the Craniosynostosis-Positional Plagiocephaly Support Group for posting our invitations to participate on their Facebook pages, to the staff of the Yale Center for Genome Analysis for expert performance of exome sequencing, and to Lynn Boyden for helpful discussions. Supported by the Yale Center for Mendelian Genomics (NIH M#UM1HG006504-05), the Maxillofacial Surgeons Foundation/ASMS (M#M156301), the NIH Medical Scientist Training Program (NIH/NIGMS T32GM007205), and the Howard Hughes Medical Institute.

## Additional information

### Funding

| Funder | Grant reference number | Author |
|---|---|---|
| National Institutes of Health | Medical Scientist Training Program, NIH/NIGMS T32GM007205 | Andrew T Timberlake Samir Zaidi |
| Howard Hughes Medical Institute | | Andrew T Timberlake Jungmin Choi Samir Zaidi Carol Nelson-Williams Erin Loring Richard P Lifton |
| American Society of Maxillofacial Surgeons | M#M156301 | Eric D Brooks John A Persing |
| Yale Center for Mendelian Genomics | NIH M#UM1HG006504-05 | Kaya Bilguvar Irina Tikhonova Shrikant Mane |

The funders had no role in study design, data collection and interpretation, or the decision to submit the work for publication.

## Author contributions

ATT, Recruited and enrolled patients, collected genetic specimens, performed SNP genotyping and Sanger sequencing, performed genetic analyses, wrote the manuscript; JC, SZ, QL, HZ, Performed genetic analyses; CN-W, Performed SNP genotyping and Sanger sequencing, performed genetic analyses; EDB, JFY, Recruited and enrolled patients, collected genetic specimens; KB, IT, SM, Directed exome sequence production; RS-M, SP, EGZ, DMS, MLD, CCD, Contributed clinical evaluations; EL, AG, KAP, Recruited and enrolled patients; CC, PWH, Collected genetic specimens; JAP, Conceived, designed, and directed study, contributed clinical evaluations; RPL, Conceived, designed, and directed the study, performed genetic analyses, wrote the manuscript

## Author ORCIDs

Andrew T Timberlake, http://orcid.org/0000-0002-8926-9692

## Ethics

Human subjects: All participants or their parents provided written informed consent to participate in a study of genetic causes of craniosynostosis in their family. Written consent was obtained for publication of patient photographs. The study protocol was approved by the Yale Human Investigation Committee Institutional Review Board.

# Additional files

### Supplementary files

• Supplementary file 1. Supplementary files for "Two locus inheritance of non-syndromic midline craniosynostosis via rare *SMAD6* and common *BMP2* alleles". (A) Exome Sequencing Quality Statistics for all members of craniosynostosis kindreds (n = 455) and autism controls (n = 3337). (B) TDT of an intergenic BMP2 risk allele and intronic BBS9 risk allele in SMAD6 mutation carriers with craniosynostosis. (C) Optimized two locus and single locus parametric models of genotype specific penetrances for SMAD6 and BMP2. (D) Family specific lod scores for each kindred under the two locus and single locus models. (E) Clinical features and BMP2 genotypes in craniosynostosis patients with rare SMAD6, SMURF1, SPRY1, or SPRY4 mutations. (F) De novo mutations identified per trio.

• Supplementary file 2. Exome sequencing quality statistics. Exome sequencing quality statistics for all members of craniosynostosis kindreds (n = 455) and autism controls (n = 3337) .

• Source code 1. R script for two locus and single locus linkage analyses.

### Major datasets

The following dataset was generated:

| Author(s) | Year | Dataset title | Dataset URL | Database, license, and accessibility information |
|---|---|---|---|---|
| Andrew T Timberlake, Jungmin Choi, Samir Zaidi, Qiongshi Lu, Carol Nelson-Williams, Eric D Brooks, Kaya Bilguvar, Irina Tikhonova, Shrikant Mane, Jenny F Yang, Rajendra Sawh-Martinez, Sarah Persing, Elizabeth G Zellner, Erin Loring, Carolyn Chuang, Amy Galm, Peter W Hashim, Derek M Steinbacher, Michael L DiLuna, Charles C Duncan, Kevin A Pelphrey, Hongyu Zhao, John A Persing, Richard P Lifton | 2016 | Yale Center for Mendelian Genomics | http://www.ncbi.nlm.nih.gov/projects/gap/cgi-bin/study.cgi?study_id=phs000744.v2.p1 | Publicly available at dbGaP (accession no: phs000744) |

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
