## [Decision Letter]

Congratulations: we are very pleased to inform you that your article, "Two locus inheritance of non-syndromic midline craniosynostosis via rare SMAD6 and common BMP2 alleles", has been accepted for publication in *eLife*. The Reviewing Editor for your submission was David Ginsburg. The additional reviewers, who have also agreed to reveal their identities were: Yuji Mishina and James Lupski.

If you have selected our "Publish on Acceptance" option, your PDF will be published within a few days; if you have opted out of the "Publish on Acceptance" option, your work will be published in about four weeks' time.

The reviewers concluded that: this manuscript reports very convincing data demonstrating epistatic interaction between rare loss of function mutations in SMAD6 and a common variant near the BMP2 locus as the genetic etiology of craniosynostosis in a subset of nonsyndromic patients. The data are of very high quality, the manuscript clearly written, and the conclusions convincingly supported by the data. The investigators performed whole exome sequencing (WES) of 132 parent-offspring trios and 59 additional probands. Seven percent of probands had damaging de novo or rare transmitted mutations in SMAD6; an inhibitor of BMP – induced osteoblast differentiation (P <10-20). Interestingly, SMAD6 mutations showed striking incomplete penetrance for the craniosynostosis trait (<60%). Genotypes of a common variant near BMP2, previously shown to be strongly associated with midline craniosynostosis, explained nearly all of the phenotypic variance in these kindred. Thus, these findings clearly documented digenic inheritance via epistatic interactions of rare and common variants as a defining feature of the most frequent cause of midline craniosynostosis. This work has very important implications for the genetic basis of other disease traits and reveals a potential mechanism for common variant genetic effects to markedly influence the manifestation of disease.

None of the reviewers had any major suggestions for required revision/correction. However, a few minor points/questions were raised, as listed below, which the reviewers all agreed could be addressed by minor changes to the text (or none), as the authors see fit.

Minor points:

1) Subsection "Exome sequencing of non-syndromic midline craniosynostosis": the authors "infer that these de novo mutations contribute to ~15% of non-syndromic midline craniosynostosis". How was this figure calculated?

2) The authors note that SMAD6 knockout mice exhibit skull abnormalities reminiscent of craniosynostosis (Estrada et al., 2011 and Retting, 2008). Is there a heterozygous phenotype in the mice? Though clearly beyond the scope of the current manuscript, it will be interesting to see whether genetic interaction between BMP2 gain-of-function and SMAD6 haploinsufficiency can be demonstrated in the mouse.

3) Though 2 previously identified GWAS hits for nonsyndromic craniosynostosis, one (BMP2) modifies SMAD6 haploinsufficiency, whereas the other (BBS9) does not, despite similar large effect sizes for both variants. Could the authors speculate about a potential similar epistatic interaction of BBS9 with rare variant in another gene(s) in a different subset of craniosynostosis patients?

4) Though most of the identified SMAD6 mutations would generate clear loss-of-function, two of them are located near the C-terminus (i.e. R465C and I490T). Have any specific functions been ascribed to this region of the molecule?

5) SMAD7 is another inhibitory Smad that affects both BMP and TGF-β pathways. Although a craniofacial phenotype is not appreciated in Smad7 mutant mice, they show similar skeletal phenotypes to Smad6 knockouts. One would think mutations in SMAD7 might also interact with the BMP2 risk allele. Does the lack of SMAD7 mutations in their patient cohort, suggest a particularly unique role for SMAD6 in midline craniosynostosis?

6) Discussion, paragraph 5: "The only other clinical finding found[…]" could be reworded as "finding present" or "observed".

---

## [Author Response]

*1) Subsection "Exome sequencing of non-syndromic midline craniosynostosis": the authors "infer that these de novo mutations contribute to ~15% of non-syndromic midline craniosynostosis". How was this figure calculated?*

As discussed in the Methods section, among 132 probands, 123 had protein-altering *de novo* mutations, compared to an expected number of 102.4. Thus ~20 more probands (~15% of all probands) had protein-altering de novo mutations than expected by chance, suggesting that at least this fraction of probands has disease owing to *de novo* mutation.

*2) The authors note that SMAD6 knockout mice exhibit skull abnormalities reminiscent of craniosynostosis (Estrada et al., 2011 and Retting, 2008). Is there a heterozygous phenotype in the mice? Though clearly beyond the scope of the current manuscript, it will be interesting to see whether genetic interaction between BMP2 gain-of-function and SMAD6 haploinsufficiency can be demonstrated in the mouse.*

There is no heterozygous phenotype described for *SMAD6* loss of function in mice. We agree that it will be interesting to determine the consequence in mice of combined *BMP2* gain of function with *SMAD6* loss of function.Titration of dosing of BMP2 will likely be important in such an experiment.

3) Though 2 previously identified GWAS hits for nonsyndromic craniosynostosis, one (BMP2) modifies SMAD6 haploinsufficiency, whereas the other (BBS9) does not, despite similar large effect sizes for both variants. Could the authors speculate about a potential similar epistatic interaction of BBS9 with rare variant in another gene(s) in a different subset of craniosynostosis patients?

This is an interesting point. We speculated in the manuscript that other common variants, particularly those with relatively large effects, may show similar epistatic interactions with rare variants. The common BBS9 risk variant, which clearly falls into this class, would be an excellent candidate for modifying risk of rare variants in another gene or genes.

4) Though most of the identified SMAD6 mutations would generate clear loss-of-function, two of them are located near the C-terminus (i.e. R465C and I490T). Have any specific functions been ascribed to this region of the molecule?

The C-terminus, which is part of the MH2 domain, is responsible for protein-protein interactions, including SMAD6’s interaction with receptor-SMADs and BMP receptors themselves. It is possible that these variants impair SMAD6 binding to these signaling molecules, impairing inhibition of BMP/SMAD signaling.

*5) SMAD7 is another inhibitory Smad that affects both BMP and TGF-beta pathways. Although a craniofacial phenotype is not appreciated in Smad7 mutant mice, they show similar skeletal phenotypes to Smad6 knockouts. One would think mutations in SMAD7 might also interact with the BMP2 risk allele. Does the lack of SMAD7 mutations in their patient cohort, suggest a particularly unique role for SMAD6 in midline craniosynostosis?*

Yes, the data would support this inference. It has been suggested that *SMAD6* is more specific to regulation of BMP signaling, whereas *SMAD7* is more specific for inhibition of TGFB-receptor - mediated SMAD signaling. Mutations in TGFBR1/2, and its downstream inhibitor SKI, have been shown to cause syndromic forms of craniosynostosis that also involve the cardiovascular system and are markedly more severe than the non-syndromic craniosynostosis phenotype resulting from SMAD6/BMP2 variants. It is thus possible that heterozygous loss of function mutations in *SMAD7* have a more extreme, potentially early lethal, phenotype than *SMAD6* mutations, accounting for why we have not seen these mutations in our cohort. Consistent with this possibility, zero heterozygous loss of function variants in *SMAD7* have been reported in the ExAC database.

6) Line 315: "The only other clinical finding found[…]" could be reworded as "finding present" or "observed”.

We agree, thanks. We have changed the sentence to, 'The only other clinical finding observed[…]'